# SimpleStrat: Diversifying Language Model Generation with Stratification

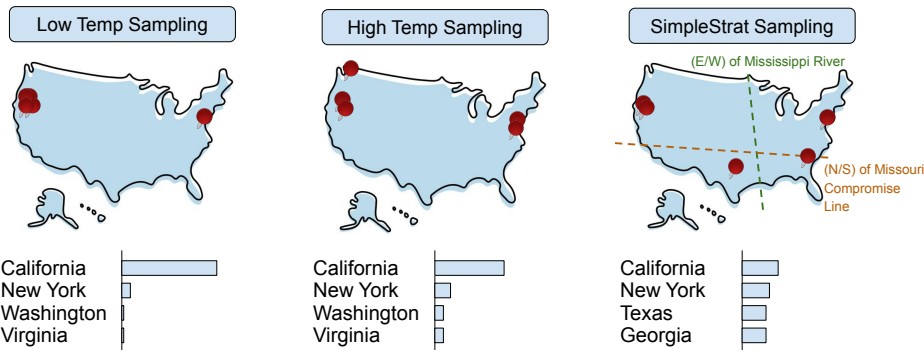

Figure 1: **Stratfied Sampling vs Temperature Scaling** Consider the LLM user request "Name a US State." SimpleStrat employs auto-stratification to utilize the LLM to identify good dimensions of diversity, for instance "East/West of the Mississippi River." Then, SimpleStrat uses stratified sampling to diversify LLM generations.

## Abstract

Generating diverse responses from large language models (LLMs) is crucial for applications such as planning/search and synthetic data generation, where diversity provides distinct answers across generations. Prior approaches rely on increasing temperature to increase diversity. However, contrary to popular belief, we show not only does this approach produce lower quality individual generations as temperature increases, but it depends on model's next-token probabilities being similar to the true distribution of answers. We propose SimpleStrat, an alternative approach that uses the language model itself to partition the space into strata. At inference, a random stratum is selected and a sample drawn from within the strata. To measure diversity, we introduce CoverageQA, a dataset of underspecified questions with multiple equally plausible answers, and assess diversity by measuring KL Divergence between the sampling distribution and uniform distribution over valid ground truth answers. As computing a posterior probability for proprietary models is infeasible, we measure recall on ground truth solutions. Our evaluation show using SimpleStrat achieves higher recall by 0.05 compared to GPT-4o and 0.36 average reduction in KL Divergence compared to Llama 3.

# 1 Introduction.

Large language models (LLMs) are routinely resampled in order to get a wide set of plausible generations. Three key settings where this is important are: 1) improving downstream accuracy with planning or search for agentic tasks (i.e. Tree-of-thought (Yao et al., 2024), AgentQ (Putta et al., 2024)), 2) estimating prediction uncertainty (Aichberger et al., 2024), and 3) generating diverse datasets for post-training (Dubey et al., 2024) and fine-tuning (Dai et al., 2023). All these use cases rely on the model generating multiple plausible generations for the same prompt when multiple answers exists.

Figure 2: **SimpleStrat workflow.** SimpleStrat employs 3 phases: 1) auto-stratification to identify good dimensions of diversity that divide the solution space into equal partitions, 2) heuristic estimation to estimate the proportion of solutions in each stratum, and 3) probabilistic prompting where a concrete prompt is randomly sampled from the prompt distribution specified by the previous two phases. Critically, diverse resampling comes from both the random choice of prompt as well as the temperature of the LLM decoding.

Naively, increasing temperature, a parameter that controllably flattens an LLM's softmax, can improve an LLM's generation diversity. However, temperature introduces two problems. First, higher temperatures degrades generation quality. Recent evidence suggests removing temperature scaling is desirable for multi-step reasoning to reduce errors compounding (Zhang et al., 2024). This is especially critical in syntax sensitive settings like code generation where low temperatures ($\leq 0.15$) are often used. Second, controlling for temperature does not necessarily improve diversity in the answer space. In Figure 1, we illustrate increasing temperature doesn't lead to meaningful increase in diversity if the model is excessively confident and suffers from mode collapse. When asked to "Name a US State," the model heavily skews towards answering "California", high temperature only marginally softens the skew while surfacing incorrect answers and hurting instruction following.

Our goal is to improve diversity when resampling LLMs, even in cases of severe mode collapse in next-token probabilities without manual intervention. Our analysis reveals that GPT-4 assigns 87% of its logit weight to "California" when prompted to name a US state. This observed bias can be attributed to the worsening of calibration due to post-training as reported in the GPT-4 tech report (OpenAI et al., 2024). This stark bias mirrors human cognitive bias, exemplified by the blue-seven phenomenon—where individuals disproportionately select blue and seven when asked to choose a random color and number. To counteract similar biases in human populations, social scientists, particularly in political polling, employ stratified sampling techniques (Simpson, 1951; Howell, 1992; Morris, 2022). We propose adapting this method to address mode collapse in LLMs.

We propose SimpleStrat, a training-free sampling approach to increase diversity. SimpleStrat improves LLM generation diversity without degradation to generation quality while ensuring that an LLM's outputs are aligned with the true distribution of answers. SimpleStrat consist of three stages: auto-stratification, heuristic estimation, and probabilistic prompting. Even if a language model cannot generate diverse solutions, we find that it can be prompted to identify useful partitions of the solution space based on the user request. We call this process *auto-stratification*. In Fig. 1, SimpleStrat identifies two semantically significant strata from user request, "Name a US State": "(East/West) of the Mississippi River" and "(North/South) of the Missouri Compromise Line."

Next, the heuristic estimation computes the joint probabilities across all strata. Back to Fig. 1, SimpleStrat then outputs the probability for all four possible regions in US. Finally, SimpleStrat samples from the joint probability distribution to augment the original user prompt with the selected stratas. We note that this approach to diversity is orthogonal to increasing temperature and hence does not affect generation quality.

We evaluate SimpleStrat on underspecified questions, specifically questions that have more than one plausible answer. However, unlike ambiguous questions more widely, an answer to an underspecified question can be easily verified to be a valid without additional context. These questions capture settings where the user is indifferent to the particular answer as long as it's valid or in settings where we wish to resample to get a set of candidates solutions. We introduce CoverageQA, a benchmark of underspecified questions with on average 28.7 equally plausible answers.

We measure diversity by computing the Kullback-Leibler (KL) Divergence from the Maximum a Posterior(MAP) distribution of answers to a uniform distribution over all valid answers. By computing the MAP using next-token probabilities, we show SimpleStrat samples from a less biased distribution.

For proprietary models where we cannot compute the Maximum a Posterior distribution, we measure the model's coverage via recall of ground-truth solutions over 100 samples. On CoverageQA, SimpleStrat leads to 0.36 reduction in KL Divergence on average on Llama 3 models and a consistent 0.05 increase in recall. We show gains on top of temperature scaling leading to improved diversity orthogonal to increasing temperature.

Concretely, our work contributes the following:

- CoverageQA dataset of 105 under-specified questions automatically generated from Wiki-Data (Vrandečić & Krötzsch, 2014) annotated with on average 28.7 valid solutions per question.
- We propose SimpleStrat a training-free approach for improving diversity with *auto-stratification* and *probabilistic prompting*.
- We demonstrate SimpleStrat improves diversity on CoverageQA with 0.36 reduction in KL Divergence on average on Llama 3 models and a consistent 0.05 increase in recall across all temperatures for GPT-4o.

## 2 RELATED WORK.

**Temperature Scaling.** Going back as far as Platt scaling (Platt, 2000) and later applied to neural networks (Hinton, 2015; Guo et al., 2017), temperature scaling controls the randomness of probability distributions[1]. For dataset generation with LLMs, Chung et al. (2023) extends temperature-based diversity by additionally downsampling previously sampled tokens. To address the decrease in quality, they advocate for human intervention to manually filter out irrelevant diversity and manually fixing wrong answers in QA tasks. We show in our work temperature scaling leaves much to be desired.

**Improving Language Model Diversity with Search.** In autoregressive generation, choices over early tokens tend to have more impact on the eventual completion. Beam search ameliorates this bias by allowing for multiple candidates in searching for the probability maximizing completion, Maximum a Posteriori (MAP) Lowerre & Reddy (1976). At the end of the search, beam search will have multiple candidate solutions encountered during search. Diverse Beam Search (DBS) proposes introducing an auxiliary dissimilarity objective quantifying the diversity among candidates in the beam (Vijayakumar et al., 2016). Especially on the task of image captioning, DBS shows improvement for discovering higher probability completions and discovering diverse continuations. Our improvements are orthogonal to beam search and our in-context approach corrects for inaccuracies in the modeled likelihoods of candidate solutions.

Other approaches (Samvelyan et al., 2024; Bradley et al., 2023) based on MAP-Elites (Mouret & Clune, 2015) require manual determined dimensions of relevant diversity and discretization of the solution space into equally-sized bins. Diversity is then achieved by mutations and evolutionary methods to cover adjacent bins. This search is potentially slow if the seed set of solutions do not already provide coverage over the solutions space. Our approach does not need seed solutions and avoids manually identifying dimensions of diversity. Instead, we rely solely on capabilities within the model.

**In-context Methods to Increase Diversity.** When LLMs were first introduced, LMs were used to augment existing datasets with more diversity (Wei & Zou, 2019; Ng et al., 2020; Dai et al., 2023). As natural language is difficult to guarantee correctness, the space of augmentations is conservatively limited to thesaurus based synonym replacement. More recently, Language Model Crossover proposes presenting a random subset of existing data points to an LLM and ask it to hallucinate more data points that likely came from the same distribution Meyerson et al. (2023). This is limited combining aspects of existing data points into new generations. Although these methods address the limitations of using the model's token probabilities by in-context learning, they are ineffective at generating meaningful diversity. They are limited to either a human identified domains of interest or trivial variations sourced from synonyms or minicking random subsets of the existing dataset.

**Applications of Diversity.** As shown by Raventós et al. (2024), dataset diversity is crucial for model generalization. Below sufficient coverage of the desired task, the model will resort to memorization,

---

[1]Use of temperature parameter goes back at least to Verhulst's development of logistic regression in response to Malthus' *An Essay on Principle of Population* (Malthus, 1798; Verhulst, 1838).

but when sufficient diversity is presented it will learn to generalize. As LLMs are increasingly used for generating synthetic data (Dubey et al., 2024), methods for diversity will be critical. This insight follows from extensive work demonstrating the benefits of data augmentation for bias mitigation (Sharmanska et al., 2020) and domain adaptation (Huang et al., 2018; Dunlap et al., 2023; Trabucco et al., 2023).

In code and math applications, checking validity efficiently enables more aggressive augmentations. One such augmentation for diversifying the languages supported by the model, data is translated to different natural or programming language (Chen et al., 2023; Cassano et al., 2023). In other domains such as images, text-to-image models have been used to do diversify data into uncommon settings. In the setting of diversifying an accumulating dataset, these methods can take advantage of an existing source of variance (for translation) or set of previously generated data points. Our primary focus is on settings where SimpleStrat is unaware of past data samples to support a wider set of applications.

**Ambiguous or Underspecified Datasets.** ClariQ (Aliannejadi et al., 2020), CLAQUA (Xu et al., 2019), and AmbigQA (Min et al., 2020) focus on assessing LM's ability to formulate clarifying questions. These question tend to have only 2 candidate solutions, as there exists a ground truth clarifying question whose answer fully specifies the question. Ambiguous Trivia QA (Kuhn et al., 2022) also looks at under-specified questions, but assume a user has contextual information that's hidden. For instance, "Where in England was she born?" or "Who was the first woman to make a solo flight across this ocean?". We distinguish our underspecified question setting in this paper as one where the user is indifferent. In this setting, the given an answer it should be easy to verify the answer is correct without additional hidden context.

Coding datasets like Description2Code (Caballero et al., 2016), Wiki2SQL (Zhong et al., 2017), SPIDER (Yu et al., 2019), code-contest (Li et al., 2022), Apps (Hendrycks et al., 2021), and Leetcode Hard Shinn et al. (2023) admit multiple valid answers. However, the space of valid implementations is infinite, making diversity difficult to measure, and good coding practices enforce preferences among valid implementations. We additionally construct CoverageQA to have an exhaustive list of ground-truth answers in order to measure the impact of diversity on coverage.

## 3 METHOD

### 3.1 WORKFLOW OVERVIEW

As illustrated in 2, SimpleStrat consist of three stages, 1) auto-stratification, 2) heuristic estimation, and 3) probabilistic prompting. For each unique user prompt, the outputs of the first two stages can be cached to avoid recomputing feed-forwards.

### 3.2 AUTO-STRATIFICATION

For a given user request, $r_{user}$, we call $S$, the space of valid solutions. In many settings, the space of potential solutions, $S$ may be naturally partitioned based on geography, parity, or demographics. The partition function, $P : S \rightarrow L$, assigns any solution $s$ from $S$ to a partition label $l_j$ in $L$ the set of partition labels. Partition functions are most useful if they're as balanced as possible. A balanced partition function minimizes $imbalance(P, L) = \max_{l \in L} (|\{s \mid P(s) = l\}|) - \min_{l \in L} (|\{s \mid P(s) = l\}|)$. The goal of auto-stratification is to search for a set of partition functions $\mathbf{P} = \{P_1, P_2, ..., P_n\}$, that are balanced. Traditionally, in settings where there are oft-overlooked or a large or infinite number of valid solutions, stratified sampling can ensure our limited budget of samples covers the space of solutions evenly.

Based on this insight, we prompt the language model to identify promising dimensions of diversity. Concretely, the language model proposes good clarifying questions that will potentially eliminate half of the potential solutions based on the user request. These clarifying questions tend to align with semantically significant differences. In the running example, when asked, "Name a US State," the states can be partitioned based on East or West of the Mississippi River. See App. C for full prompt.

### 3.3 Heuristic estimation

As previously observed in Zou et al. (2022); Yan et al. (2023); Halawi et al. (2024), LLMs can used in forecasting to estimate well-calibrated probabilities of events that have not yet occured. For forecasting, the model success benefits substantially from having updated news through web search. Although our unnecessary for the offline benchmarks we consider, this may be helpful for accurate estimation depending on the application. However, as our goal is diversity, we stand to benefit even from coarse-grain approximate proportions. We employ a similar reasoning template as Halawi et al. (2024) to estimate the proportion of valid solutions lie within each strata.

In heuristic estimation, we look to estimate the joint distribution for each stratum, $\vec{l} = [l_1, l_2, l_3, ...]$. Formally, we define the weighted-stratification as $\mathcal{W} = (\mathbf{P}, \rho)$, where $\rho(\vec{l}) = Pr_{s \sim S}[P_1(s) = l_{1,j}, P_2(s) = l_{2,j}, P_3(s) = l_{1,j}, ...]$ for $\mathbf{P}$ identified in auto-stratification. To improve scalability, we assume the partition functions are independent and multiply the marginal probabilities to get the joint probabilities associated with each stratum.

$$\rho(l_1, l_2, ..., l_m) = \prod_i \rho_i(l_i) \tag{1}$$

We ask the LLM for each $l_j$, to estimate the marginal proportion of solutions this holds for. As this may not add up to 1, we normalize the estimates to form a proper probability distribution. For simplicity, we focus in this work on settings where all solution in the solution space is equally like. As noted in Sec. 3.2, we encourage the LLM to propose balanced partitions. However, heuristic estimation allow us to support imbalanced partitions by reweighing the sampling to favor stratum with more potential solutions. More details on prompting in App. D. In Fig. 2, the LLM determines the joint probabilities across two stratas, the Mississippi River and the Missouri Compromise Line.

### 3.4 Probabilistic Prompting.

Post heuristic estimation, a set of statum is sampled from the joint probability distribution in Eqn. 1. This implicitly forms a *probabilistic prompt*, which specifies a distribution over concrete language model prompts. After a prompt is sampled, the LLM is then used to sample from within the stratum. Back to Fig. 2, East and South are sampled from the Mississippi and Missouri strata respectively, augmenting the final prompt with diverse specifications.

Formally, call $\vec{l}$ a stratum defined by choices of $l_{i,j}$ for each $P_i$ across all $i$. Call $Prompt$ a function that maps the stratum, $\vec{l}$ to a concrete prompt, $Prompt(\vec{l})$. Intuitively, the probabilities of the prompt distribution is defined by $Pr[Prompt(\vec{l}) = \rho(\vec{l})$. We can then compute the probability of a solution

$$Pr[s] = \sum_{\vec{l}} Pr[\text{Prompt}(\vec{l})] * Pr[s \mid \text{Prompt}(\vec{l})] = \sum_{\vec{l}} \rho(\vec{l}) * Pr[s \mid \text{Prompt}(\vec{l})] \tag{2}$$

The specific language model's next-token probabilities define $Pr[s \mid \text{Prompt}(\vec{l})]$.

As the probabilistic prompt is human readable form, the user can inspect the properties and the proportions and modify it to adjust for unwanted bias or remove unwanted factors. For instance, when proposing English baby names, we may want the model to propose male vs female names equally often, even though there are more female than male baby names [2]. This interpretability and controllability is a major advantage of SimpleStrat in practice.

## 4 CoverageQA Dataset

### 4.1 Overview

We wish to evaluate generation diversity in settings where 1) user requests have more than one distinct correct answer, 2) and answers are equally likely, and 3) answers do not require hidden or implicit

---

[2]As reported in Wilson (2016), there are 18,993 unique names for girls and 13,959 for boys in 2015 report by Social Security Administration.

context to verify. These three properties allow us to measure diversity in the sense of whether the language model will provide coverage over the full solution space when resampled. Unfortunately existing benchmarks discussed in Sec. 2 do not satisfy these properties. We introduce CoverageQA to assess the language model generation diversity. The dataset consists of two splits: CoverageQA-Curated, manually curated naturally underspecified questions, and CoverageQA-Wikipedia, and auto-generated dataset of underspecified questions.

## 4.2 COVERAGEQA-WIKIPEDIA APPROACH

To generate CoverageQA-Wikipedia, we leverage the Wikidata knowledge base which contains all relational mappings between entities and properties in Wikipedia. Our generation process starts with an initial item-property pairing and a constraint on the number of correct answers. We then perform a recursive search through Wikidata to find all sets of item-property constraints and their corresponding answers that meet our criteria. These constraints are subsequently transformed into natural language questions using GPT-4.

Consider an initial pairing of the Wikidata item "country" with the property "instance of". We might specify that we want between 20 and 40 valid answers. Our search would then yield a set of all constraints from the knowledge base that fit the initial conditions, such as "instance of country, located in Europe, uses Euro as currency". GPT-4 would convert this into a natural language question like "Name a country located in Europe that uses the Euro as its currency."

This approach has several advantages: 1) it allows us to create a diverse and extensive benchmark that can be easily updated with weekly updates to Wikidata, 2) it allows us to arbitrarily specify the size of the solution space as constraints can be added or removed to form; and 3) this process in principle can curate a large dataset with little manual effort or supervision. In the initial instantiation of CoverageQA dataset, we publish 105 questions across 4 domains, each corresponding to a different initial seed item-property pair. To ensure quality, we employ both automatic filters (e.g., excluding certain generic properties) and manual curation to remove redundant or unsuitable questions. This dataset can be substantially expanded as we only used 4 domains, but we leave this for future work. For a details on the dataset breakdown and details on the question generation process, please refer to Appendix A.1.

## 5 RESULTS

### 5.1 EVALUATION SETUP

For the primary empirical evaluation of SimpleStrat, we use gpt4o-2024-08-06. To obtain true answer distributions and perform divergence from uniform analysis, we use open-source models from the Llama 3 and 3.1 families, specifically the 8B and 70B variants. The inference of these models were run on 8 A100-80GB GPUs. Additionally, we leverage claude-3.5-sonnet-20240620 for baseline evaluations on two datasets: CoverageQA Curated and CoverageQA Wikipedia. For CoverageQA, we used WikiData version from 07-03-2024.

### 5.2 MEASURING DIVERSITY

We consider two measures of diversity. For models with accessible softmax next-token probabilities, we compute the a Posterior probability of each solution in the solution space. We then define distributional diversity as the distributional distance between the a Posterior sampling distribution and the ground-truth distribution derived from these probabilities. For CoverageQA, the ground-truth distribution is uniform over valid solutions and zero elsewhere. In general, it can be more complex.

In setting where we do not have access to the next-token distribution, we evaluate diversity by resampling responses to CoverageQA 100 times per question. This allows us to empirically observe the diversity in the form of coverage. We call this coverage diversity. To measure coverage, we report the recall ($unique\ valid\ solutions/total\ unique\ valid\ solutions$) on the reference solutions. Note

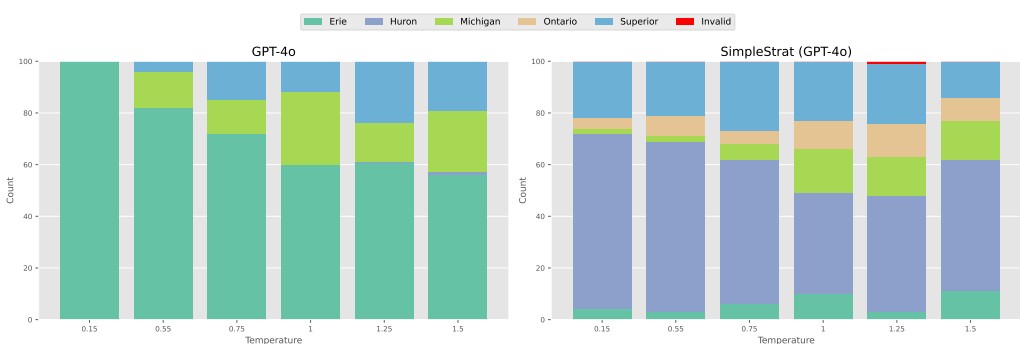

Figure 3: **Diversity scaled with temperature.** We show 100 resamples of "Name one Great Lake in the United States." On the right, we show the result of resampling GPT-4o 100 times per temperature. In contrast to SimpleStrat on the left, GPT-4o at temperature 1.5 still only samples Lake Huron once and never samples Lake Ontario. SimpleStrat improves the diversity across all temperatures.

that this is an unconventional measure of recall as it's over the solution space[3]. To ensure this doesn't come at the cost of quality, we also show precision is not reduced.

### 5.3 QUALITATIVE EXAMPLE

Consider the question "Name one Great Lake in the United States." as shown in Fig. 3. We see that temperature scaling with GPT-4o results in a strong preference/bias for Lake Erie. This is certainly a correct continuation and under the language modeling objective should be incentivized. Increasing the temperature helps sample the next most likely candidate solutions more often. However, even when increasing the temperature past 1 there is still low coverage over the solutions space. Specifically, Huron is only seen once out of 100 samples at 1.5 temperature, and Lake Ontario is never observed. This is undesirable if the data is used to proposing candidate plans, generating test cases, or generating training data. Not only is there insufficient coverage over all possible solutions, but the model consistently has a strong preference for Lake Erie. This undesired biases in generations may lead to problems in downstream use cases.

In Fig. 3, we observe a much more uniform distribution over valid solutions when using SimpleStrat. Notably, we observe full coverage over all 5 Great Lakes. At lower temperatures, there is still a preference of a single lake over the others, in this case Lake Huron. However, this is less pronounced at higher temperatures and is a significant improvement over GPT-4o without SimpleStrat.

### 5.4 COVERAGE DIVERSITY ON PROPRIETARY MODELS

We first assess coverage diversity, specifically, the model's ability to recall all the valid solutions upon resampling. This measure is clearly impacted by temperature as temperature zero or greedy decoding of LLMs leads to a single deterministic result. We compare the coverage diversity (recall) of SimpleStrat, GPT-4o, and Claude 3.5 Sonnet as a function of temperature. We sweep over temperatures from 0.15 to 1.5. Although not shown in the evaluation, note that SimpleStrat has the advantage of providing diversity even when sampled at temperature zero. SimpleStrat with GPT-4o leads to an improvement to recall across all temperatures as shown in Fig.4. On the CoverageQA-Curated, we see a consistent 0.2 increase in recall over the same base model, GPT-4o. On CoverageQA-Wikipedia, we see as much as 0.05 increase in recall

Our quantitative results are consistent with the Great Lakes example in Fig. 3. Scaling temperature alone does not lead to as much coverage diversity as combining with SimpleStrat. The recall importantly does not come at the expense of quality as measure by precision as shown in App. B.

---

[3]Not to be confused with conventional recall where we might measure how many valid solutions the LLM recognize as valid.

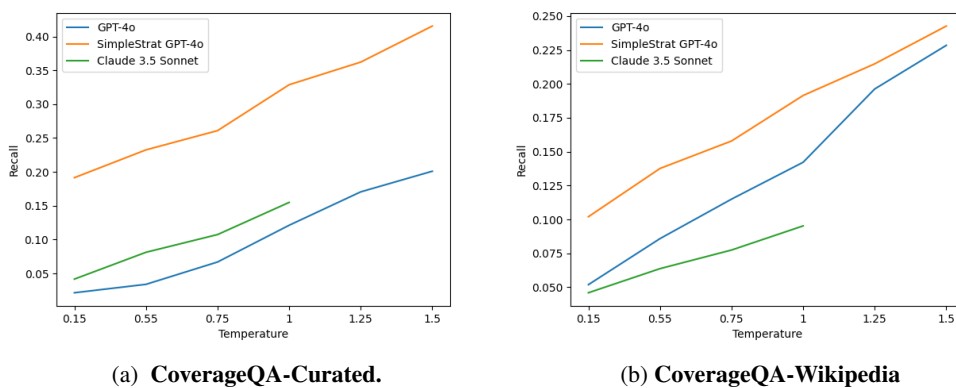

(a) **CoverageQA-Curated.**  (b) **CoverageQA-Wikipedia**

Figure 4: **Diversity measured with recall scaled with temperature.** The figure shows the improved recall on CoverageQA compared to GPT-4o and Claude 3.5[4]. Recall indicates the percentage of ground truth questions observed after sampling 100 times. The benefit of SimpleStrat is especially pronounced at low temperatures, but the benefit is evident across all temperatures.

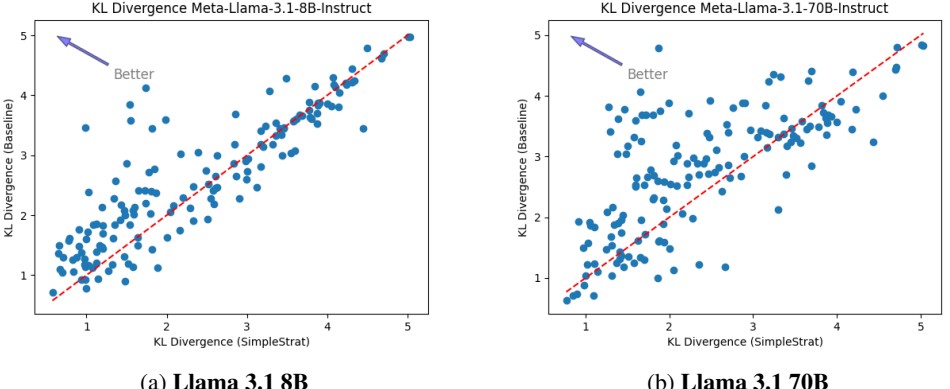

(a) **Llama 3.1 8B**  (b) **Llama 3.1 70B**

Figure 5: KL divergence from uniform for Baseline vs SimpleStrat on CoverageQA Wikipedia. Lower divergence indicates closer alignment with the desired uniform distribution, arrow indicates direction of maximum improvement from baseline

## 5.5 DISTRIBUTIONAL DIVERSITY WITH LLAMA 3

We use the Llama 3 model family to analyze answer distributional diversity in CoverageQA. Open-source models let us calculate exact expected distributions by examining logits of all valid continuations for a prompt. This is not possible with GPT-4o and Claude 3.5 Sonnet, where reliably estimating the true probabilities would require extensive sampling. Our approach efficiently determines true expected distributions of valid answers for any input, improving analysis accuracy, and overcoming the resource constraints of high count sampling.

For our baseline, we prompt the models and directly compute $Pr[s|Prompt(\vec{l})]$ for each solution, $s$. This is simply the product of the individual next-token probabilities. For SimpleStrat, the probability involves the next-token probability conditioned on the prompt weighted by the probability the prompt is selected. Concretely, the probability an answer is sampled by SimpleStrat can be computed based on Eqn.2. The next-token probability based posterior $Pr[s|Prompt(\vec{l})]$ computed just as the baseline, and we do a sum weighted by the joint probabilities assigned in heuristic estimation. We assign remaining probability density to an "Invalid" category to form a proper distribution. The probabilistic formulation allows us to easily compute the a Posterior distribution of SimpleStrat.

---

[4]Claude does not allow for temperatures above 1.

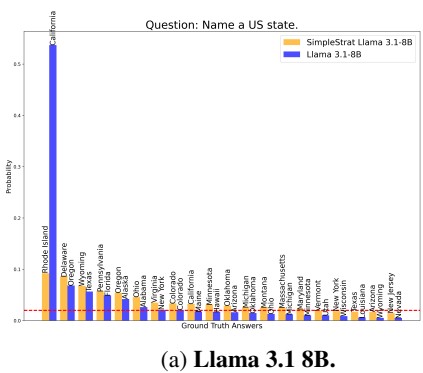 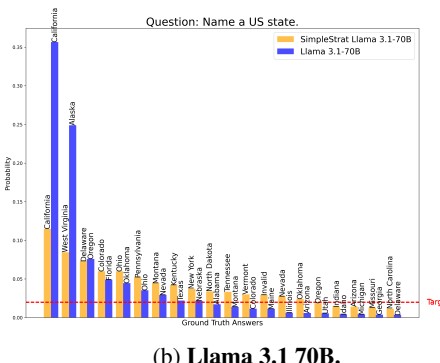

| (a) **Llama 3.1 8B.** | (b) **Llama 3.1 70B.** |

Figure 6: **Distributional Diversity Comparison.** We show the a posterior probability as defined by next-token-probabilities for ground truth answers on Llama 3.1. For both 8B and 70B, SimpleStrat provides meaningful improvement to the sampling distribution both for values previously over-represented in the distribution and those previously underrepresented.

Table 1: Comparison of Baseline and SimpleStrat Average KL Divergence for Different Models on CoverageQA-Curated and CoverageQA-Wikipedia. Smaller numbers reflect closer alignment to uniform distribution.

| **Model** | **CoverageQA Curated** | | **CoverageQA Wikipedia** | |
|---|---|---|---|---|
| | **Baseline** | **SimpleStrat** | **Baseline** | **SimpleStrat** |
| Meta-Llama-3-8B-Instruct | 2.78 | **1.74** | 2.75 | **2.47** |
| Meta-Llama-3.1-8B-Instruct | 2.47 | **1.19** | 2.60 | **2.39** |
| Meta-Llama-3-70B-Instruct | 3.24 | **2.17** | 3.28 | **2.73** |
| Meta-Llama-3.1-70B-Instruct | 2.70 | **1.54** | 2.78 | **2.38** |

Our ground-truth distribution is a distribution where each valid answer has equal probability, and no invalid answers have probability density. We use Kullback–Leibler (KL) Divergence as the discrepancy metric to measure how much our distribution deviates from a uniform distribution. Across the four models in the Llama 3 family, SimpleStrat achieves an average reduction in KL divergence from uniform of 1.14 compared to the baseline on the curated CoverageQA dataset. For the general CoverageQA dataset, the reduction is 0.36. These results indicate that SimpleStrat produces a sampling distribution much closer to uniform than the baseline method.

Additionally, we analyze the KL divergence on a per-question basis to identify where our method achieves improvement (a more uniform distribution). The scatter plot in Fig 5b shows KL divergence values for SimpleStrat (y-axis) versus the baseline (x-axis) for each question in the CoverageQA Wikipedia dataset. Points above the diagonal line represent questions where SimpleStrat outperforms the baseline by yielding a lower KL divergence. The vast majority of points fall on or above this line, indicating that SimpleStrat consistently produces improvement in generating more uniform distributions across the questions in the dataset.

In Fig. 6, we see an example of the a posterior distribution for Llama 3.1 with and without the correction due to SimpleStrat. The base model distribution is strongly biased towards it's most desired output. As illustrated in Fig. 1, California is preferred by a long margin. Thus, it's not surprising that we observed little diversity when by simply increasing temperature. In contrast, SimpleStrat provides a much more uniform distribution. The over represented solutions are adjusted to be lower and the under represented solutions are adjusted to be higher. We note that the distributions are still not perfectly uniform. For more examples, see App. E. Although Fig. 6 shows the larger model has a closer to ideal distribution, the results in Table 1 indicate larger models have on average worse diversity. This may be a result of memorization exacerbated in larger models.

**Model versions.** Interestingly, Llama 3.1 handles underspecified questions better as it has consistent lower KL Divergence. Although we do not have access to the training methodology, the likely increase in dataset size and more careful model design seems to improve based mode capabilities on underspecified questions.

## 6    LIMITATIONS

Although SimpleStrat shows improvement empirically, it is sensitive to the model selecting good meaningful axis in *auto-stratification* and correct joint probabilities in heuristic estimation. As work on LLMs for forecasting improve, we expect LLMs to produce better estimates especially when given access to external data and data analysis tools. For our prototype, we restricted to studying the model's intrinsic capabilities. Further, the model may have biases concerning race and gender that may be reflected also in the auto-stratification and heuristic estimation. As such, it is recommended the probabilistic prompt distribution of SimpleStrat is carefully inspected for critical applications. Finally, CoverageQA is a dataset of short responses. Although this make evaluation more practical, we believe SimpleStrat will be especially impactful in settings that require low temperature especially multi-step reasoning as identified by Zhang et al. (2024).

## 7    CONCLUSION

In this paper, we propose SimpleStrat which offers an innovative alternative by leveraging the LLM itself to partition the solution space into distinct strata. This process we call *auto-stratification*. At inference time, a random stratum is selected, and a sample is drawn from within that stratum. This approach achieves better diversity without sacrificing quality as increasing temperature would.

To quantitatively measure diversity, we introduced the CoverageQA dataset, which consists of underspecified questions with multiple equally valid answers. We measure diversity with two metrics: for open-source models, we measure distributional difference with KL Divergence and for proprietary models, we measure coverage over the set of ground-truth solutions. Our rigorous evaluation on both proprietary and open-source LLMs demonstrated that SimpleStrat achieves significantly higher recall and produces answer distributions closer to uniform compared to traditional temperature-based sampling methods.

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

# A APPENDIX

## A.1 COVERAGEQA DATASET

**Generation Procedure**

To generate the questions, we manually came up with initial item and property pairings to run the recursive search. We constrain the recursive search to yield between 20-40 possible answers to keep the questions within common and relevant categories. We found that with fewer than 20 answers, the questions become too obvious, while with more than 40, they tend to get too specific and stray from general knowledge. The recursive search first finds all items that satisfy the initial conditions, then iteratively adds properties in steps until either the maximum depth (number of constraints) is reached or the number of answers falls outside the desired range. We blacklist properties that are detrimental to high-quality question generation, such as an item's presence in a specific database, numeric properties like population, and properties that introduce high ambiguity. We then manually evaluate the generated conditions and answers to ensure they meet our criteria. With an appropriate initial condition, one query can generate hundreds of valid constraints that can later be turned into questions. Finally, we use GPT-4 to convert these constraints into natural language.

# B F1 SCORES

We show F1 scores in Fig. 7 to emphasize that the precision does not change substantially as a result of our method. Precision is calculated over the set of 100 attempts how many are in the ground truth. Recall as mentioned is calculated as how many unique ground truth solutions were observed in the 100 attempts.

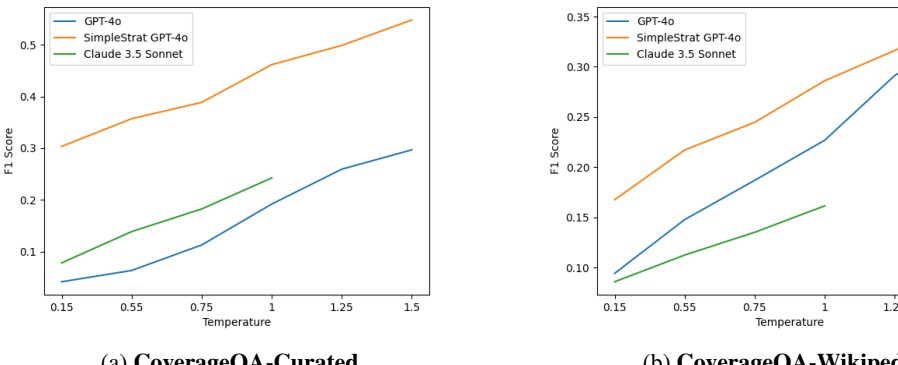

(a) **CoverageQA-Curated**          (b) **CoverageQA-Wikipedia**

Figure 7: **F1 score scaled with temperature.** The figure shows similar curves to recall on CoverageQA. This indicates the improved diversity does not come at a cost to precision.

Table 2: CoverageQA Domains

| Domain | Question Count | Average Number of Answers |
|---|---|---|
| General Knowledge (Curated) | 10 | 64.1 |
| US National Parks | 6 | 12.2 |
| Geography Questions | 75 | 27.4 |
| Chemical Elements | 14 | 17.5 |

# C AUTO-STRATIFICATION PROMPT

We provide the full prompt in Tbl. 3. To improve prompt adherence, we provide one in context example in the form of one simulated round of multi-turn conversation, i.e. we provide an example set of reasoning following the template.

**System Prompt:**
You're a helpful brainstorming assistant that is careful to consider all factors to a problem.

**User:**
I am tasked with the following request:
*% User Request*
Help me brainstorm how to respond to the user request by providing a list of True/False properties the solution may or may not have. Use the following step-by-step to come up with good properties:

1. If you were playing 20 questions, what's a good first question to ask that would split the possibilities in half?

   List at least 5 questions and their corresponding properties.

   Question: <Description>

2. Rewrite each question as a True/False property that's true for one half and false for the other.

   Question: <Description>

   True/False Property: <Property Description>

3. For each property, come up with an example that would satisfy the property.

   Property: <Description>

   Example: <Description>

   Is it a valid answer to the user's request? <Yes/No>

4. For each property, come up with an example that would not satisfy the property.

   Property: <Description>

   Example: <Description>

   Is it a valid answer to the user's request? <Yes/No>

5. Does the property mention a candidate answer in it?

   Property: <Description>

   Does the property mention a candidate answer in it? <Yes/No>

6. For each property, list whether we should include it or not in the final list of properties. Do not include ones where an example from above is not valid or if it mentions a candidate answer in it.

   Property: <Description>

   Include in final list? <Yes/No>

Final List of True/False Properties:

1. <Property Description 1>
2. <Property Description 2>

Ensure all properties are listed are sentences that are either True or False

Table 3: Full prompt for Auto-stratification.

## D    HEURISTIC ESTIMATION PROMPT

We first take each partition function from auto-stratification and estimate a starting probability with the prompt in Table 4. This prompt is heavily inspired by Halawi et al. (2024). We then collect all the proportions and pass it through a final Heuristic Estimation prompt to remove redundant properties (negations for instance) and give the model a chance to correct any incorrect probabilities. See Table 5 for full prompt. Finally, we ask the model to select at most 3.

Note that for performance reasons, we estimate the marginal probabilities and make a simplifying assumption of independence. This is not strictly true if one partition function is the negation of the other. This leads potential stratum assigned positive probability but actually the stratum has no solutions. Otherwise, there would be $2^{\text{\# of Partition Functions}}$ strata to estimate probabilities of. Further, LLMs seem less reliable when asked to estimate fine-grained probabilities, whereas most marginal probabilities are by design close to 0.5.

Formally, if $P = \neg Q$, the the stratum $P \wedge Q$ has zero probability, even though we assumed it to be $Pr[P] * Pr[Q]$. We handle approximation error in estimating the true prompt distribution by

**System Prompt:**
You are an expert superforecaster, familiar with the work of Tetlock and others. Your mission is to generate accurate predictions for forecasting questions. Aggregate the information provided by the user. Make sure to give detailed reasoning.

**User:**
I am tasked to estimate the probability that a random solution to "*User Request*" has the following property "*Partitioning Property*"
Instructions:

1. Provide at least 3 reasons why the answer might be no.

   { Insert your thoughts }

2. Provide at least 3 reasons why the answer might be yes.

   { Insert your thoughts }

3. Rate the strength of each of the reasons given in the last two responses. Think like a superforecaster (e.g. Nate Silver).

   { Insert your rating of the strength of each reason }

4. Aggregate your considerations.

   { Insert your aggregated considerations }

5. Output your answer (a number between 0 and 1) with an asterisk at the beginning and end of the decimal.

   { Insert your answer }

Table 4: Prompt for Partition-specific Heuristic Estimation.

---

**System Prompt:**
You are an expert superforecaster, familiar with the work of Tetlock and others. Your mission is to generate accurate predictions for forecasting questions. Aggregate the information provided by the user. Make sure to give detailed reasoning.

**User:**
I'm playing a game where my friend has been tasked to:
"*User Request*"
I have the following Y/N statements I can ask my friend. I have probabilities that I think it's true: % Insert numbered list of partitions and proportions.
Instructions:

1. For each Y/N statement, is it redundant with another statement?

   Y/N statement: <description>

   Is redundant? <Y/N: Explanation>

2. Are any of the probabilities in accurate? If it's sufficiently accurate just report back the same value.

   Y/N statement: <Description>

   Is accurate? <Y/N: Explanation>

   Probability: <Probability>

3. Pick at most three statements that are least redundant and pair well together. Prefer ones that are closest to 50% for most information.

Final List of True/False Properties:

1. <Y/N Properties> :: <Probability>
2. <Y/N Properties> :: <Probability>

Table 5: Prompt for Final Heuristic Estimation.

---

allowing the model to reply "Invalid" to trigger a resample. With this adjustment, the probabilistic prompt distribution is maintained for this extreme case. This correction however does not ameliorate potential issues with

# E  ADDITIONAL PLOTS: DISTRIBUTIONAL ANALYSIS WITH LLAMA

We provide additional examples in Fig 9 and scatter plots for Llama 3 in Fig 8

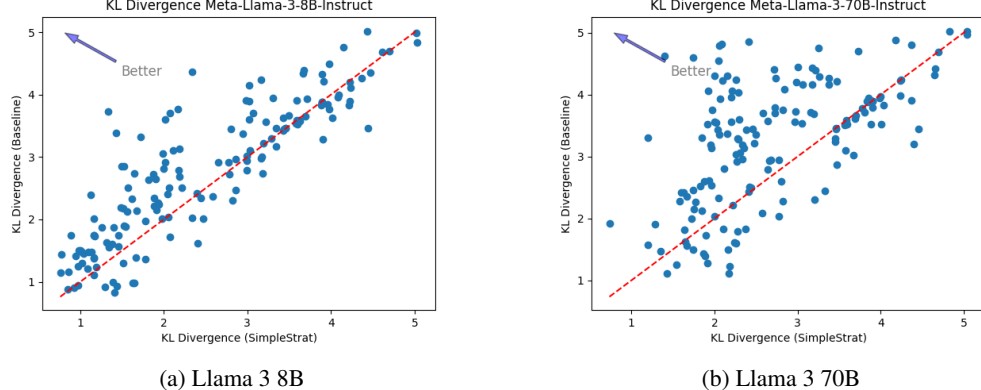

(a) Llama 3 8B                    (b) Llama 3 70B

Figure 8: KL divergence from uniform for Baseline vs SimpleStrat on CoverageQA Wikipedia. Additional plots for Llama 3 8B and 70B models

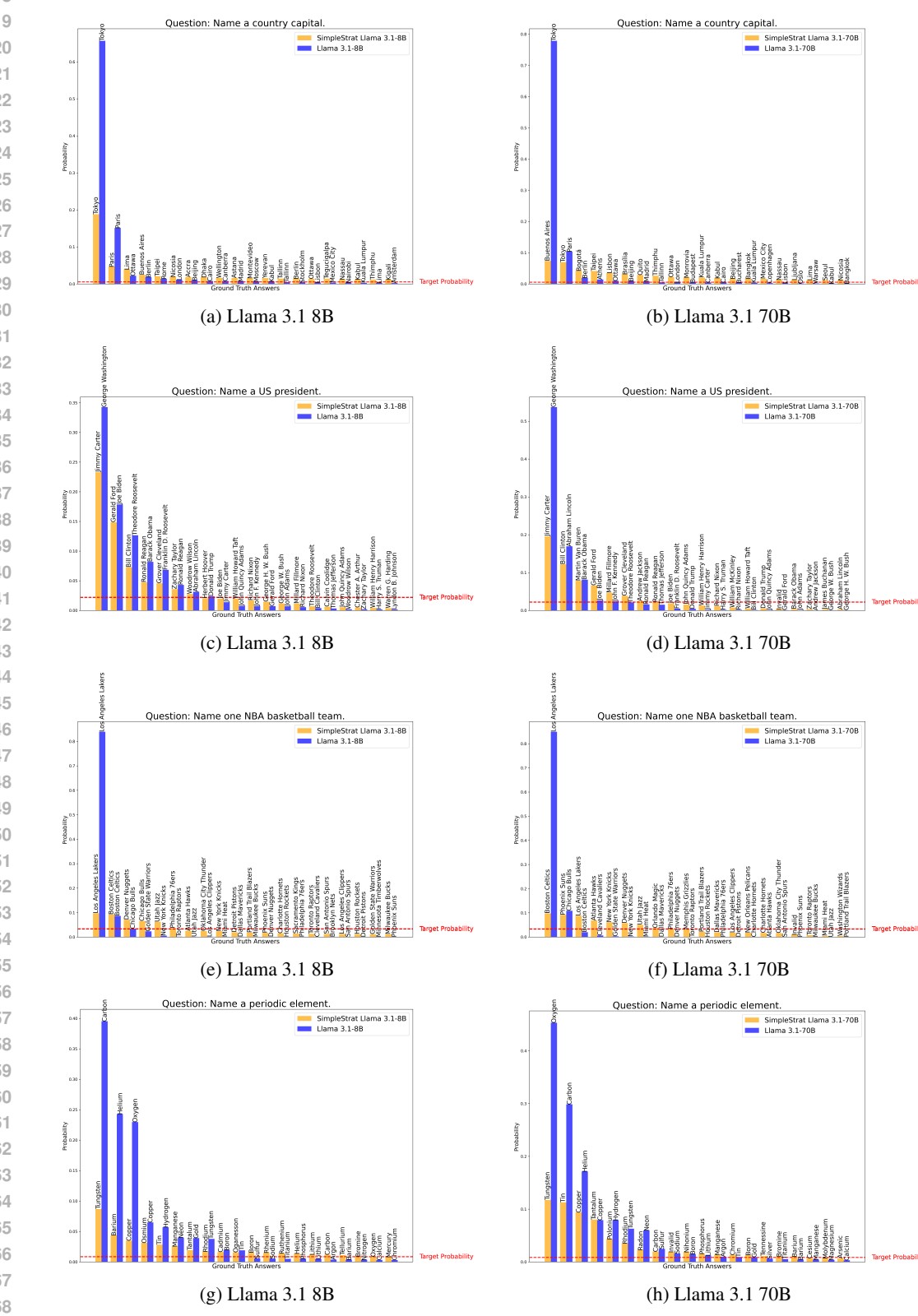

Figure 9: **Baseline vs SimpleStrat Probability Distributions** This figure shows the answer distributions for 4 additional questions from CoverageQA curated. Each row represents a different question, showing distributions for Llama 3.1 8B and 70B.

