# OpenReview forum: "SimpleStrat: Diversifying Language Model Generation with Stratification"
_ICLR.cc/2025/Conference — Submitted to ICLR 2025_

### Official Review · Reviewer_5Rwv · 2024-10-20

**Soundness:** 2
**Presentation:** 2
**Contribution:** 2
**Rating:** 5
**Confidence:** 3

**Summary:**

This paper introduces SimpleStrat, a training-free sampling method for increased diversity of LLM generation, and validates its effectiveness on a self-constructed dataset named CoverageQA, containing 105 under-specified questions. Both the method and dataset contribute to the research community.

**Strengths:**

1. The motivation, splitting the answer space into subspace (i.e., strata) along the automatically identified dimensions, is intuitively effective for higher diversity.
2. The diagram and examples clearly illustrate the main concept of the core idea.

**Weaknesses:**

1. Despite the intuitiveness,  the motivation needs to be verified, at least empirically, for broader impact. Otherwise, this paper is limited to a method and a small dataset without inspiring insights, leading to limited contribution.
2. Despite the verified effectiveness, the proposed method is only verified on a small dataset. The generalization and robustness is not clear. This makes the paper more like a prototype without substantial validation.
3. Despite the drawbacks of temperature sampling appointed by the paper, temperature sampling is still used in SimpleStrat. At least, a lower temperature in SimpleStrat should be demonstrated.
4. The writing still needs to be further polished, like:
-
  1. l046, l124: delete the final period.
  2. l102: grammar error for "verified to be a valid without"
  3. l135: citep for "Lowerre & Reddy (1976)."
  4. l181: grammar error for "the given an answer it"
  5. l196: grammar error for "As illustrated in 2, SimpleStrat consist of three stages"
  6. l218: grammar error for "LLMs can used in"
  7. l337: wrong statement for "On the right"
  8. l351: grammar error for "increasing the temperature past 1 there is"
  9. l236: Now that "For simplicity, we focus in this work on settings where all solution in the solution space is equally like.", Heuristic Estimation is not used in the main experiments, and better to be moved from a separate subsection to a discussion section.

**Questions:**

1. Could you please provide more evidence of lower temperature for SimpleStrat? Please refer to the 3rd point in the weaknesses.
2. Based on my understanding, "coverage diversity" measures the recall of valid solutions, reflecting the stability of LLM generation instead of diversity.  Could you please interpret more of its rational?
3. Based on Figure 3, the answer distribution of SimpleStrat is far from uniform distribution, diveraging from the assumption. This seems to indicate the ineffectiveness of stratified sampling. Could you please provide some clarification?
4. l424: I'm confused about the relationship between "the product of the individual next-token probabilities" and diversity. At least, length-normalization should be used to eliminate the effects of answer lengths.
5. KL divergence only indicates uniformality among all the possible solutions provided by the model, rather than within the whole solution space. This makes it unsuitable to measure the diversity in solution space.
6. This method needs to generate strata dimensions for each prompt.
-
  1. For generations with few tokens, it's not clear if this method is more effective than temperature sampling at the same token cost.
  2. For generations with many tokens, it's not clear whether an easy dimension partition still exists.

---

### Official Review · Reviewer_391L · 2024-10-29

**Soundness:** 2
**Presentation:** 3
**Contribution:** 2
**Rating:** 3
**Confidence:** 4

**Summary:**

Generation diversity of LLMs is an important research areas for several reasons such as better downstream accuracy, generate diverse datasets for post-training, RLHF, etc. Traditional approaches for diversifying include temperature scaling, beam search, in-context learning, etc. However, those methods have their own drawbacks such as lower generation quality, limited diversity improvement, etc. This work proposes a new approach for improving diversity called SimpleStrat. SimpleStrat uses the language model itself for diversity improvement. It includes three stages: 1) auto-stratification, 2) heuristic estimation, 3) probabilistic prompting. Auto-stratification prompts the LM to identify promising dimensions of diversity. Heuristic estimation estimates the joint distribution for each stratum. At probabilistic prompting stage, a set of stratum is sampled and a probabilistic prompt is formed. After a prompt is sampled from the probabilistic prompt, it's used as input into LLM for generation.
To evaluate the proposed approach, CoverageQA dataset is adopted which includes two splits: 1) CoverageQA and CoverageQA-Wikipedia. To measure diversity of the generation, it computes the KL divergence between MAP distribution of answers to a uniform distribution over all valid answers. For models that MAP distribution can't be calculated, it measures the model's coverage via recall of ground-truth solutiosn over 100 samples.
The main contributions of this work include: 1) a dataset of under-specified questions, 2) the SimpleStrat method, 3) the experimental results show that the proposed method improves diversity, specifically 0.36 KL divergence reduction on average and 0.05 increase in recall.

**Strengths:**

1) One of the contribution of this paper is the creation of a dataset CoverageQA for diversity measurement, which can benefits this area as there're not many existing datasets.
2) The proposed method is interesting and leverages LLM itself for diversity improvements. One of the advantages of this idea is that with the improvement of the LLM per se, the generation diversity can also be improved.
3) The proposed approach is evalauted on both open-sourced and proprietary models, and the experimental results show its efficacy in diversity improvements.

**Weaknesses:**

1) For evaluation, this work mostly focuses on underspecified question, however, it's not clear how this can impacts the model performance on questions with one answer.
2) The experiments consists of one baseline and this baseline is not LLM based. There're several LLM based approach. It would be better to compare the proposed approach with them.
3) How does this approach impact accuracy? This question is not discussed and answered in the work.
4) One question I have for the proposed approach is how applicable it is to other areas besides question answering such as code generation, agents, etc. One relevant question is that [the Figma AI tool keeps generating the same weather app without diversity](https://siliconangle.com/2024/07/02/figma-disables-new-ai-tool-repeatedly-cloned-apples-weather-app/). Is it possible to use the proposed approach for solving this problem?

**Questions:**

Please see the weaknesses.

---

### Official Review · Reviewer_QDaC · 2024-11-04

**Soundness:** 3
**Presentation:** 2
**Contribution:** 2
**Rating:** 3
**Confidence:** 4

**Summary:**

This paper presents SimpleStrat to increase the diversity of LLM generation without hurting the quality. SimpleStrat is a prompting method, consisting of three stages: auto-stratification, heuristic estimation, and probabilistic prompting. The rough idea is to first prompt LLM itself to produce conditions that split the generation space into different partitions, and then combine user query with the condition for the final generation such that the output follows a given partition. For evaluation, the authors propose CoverageQA. Experiments show SimpleStrat achieves good performance.

**Strengths:**

* The proposal of SimpleStrat, an interesting idea for increasing generation diversity.
* CoverageQA, a simple evaluation benchmark for LLM generation diversity.

**Weaknesses:**

* Lack of ablation
* Evaluation is weak

**Questions:**

1. SimpleStrat includes three stages but there is no analysis about how each stage affects the generation performance. What’s the quality of auto-stratification? What if the generated stratification is wrong? Besides, heuristic estimation seems unnecessary: we can always randomly sample a strat and use it for prompting. No ablations provided throughout the paper.
2. In the first stage, we could produce multiple dimensions for the stratification. Does the number of dimensions matter? How would it affect the performance?
3. Apart from the KL divergence and recall, what’s the precision of model’s generation? There is no direct evaluation about the model's quality although the authors claim SimpleStrat doesn’t hurt quality.
4. What’s the unique rate for a model's generation over the 100 samples? i.e. how many unique samples?
5. CoverageQA seems not very practical. What if applying SimpleStrat to more practical tasks, such as reasoning? It’s common to sample multiple responses in resonsing tasks and report pass@K? would SimpleStrat be better in such problems?

---

### Meta-Review · Area_Chair_9XJV · 2024-12-20

**Metareview:**

Based on the reviewers' feedback, I recommend not to accept this paper at this time. While the paper presents an interesting approach to improving LLM generation diversity through SimpleStrat, the reviewers identified several evaluation limitations that need to be addressed: the small dataset focused only on underspecified questions, insufficient comparisons with existing LLM-based diversity methods, and concerns about the appropriateness of the evaluation metrics. The reviewers also noted the lack of analysis on diversity-quality tradeoffs and missing ablation studies. Without author response to these methodological concerns, the paper's claims remain insufficiently validated.

**Additional Comments On Reviewer Discussion:**

I have read the messages in the discussion period and my opinion has been summarized as in the metareview above. I considered these points in my recommendation.

---

### Decision · Program_Chairs · 2025-01-22

Reject